# The Role of Adipokines in Health and Disease

**DOI:** 10.3390/biomedicines11051290

**Published:** 2023-04-27

**Authors:** Vicente Javier Clemente-Suárez, Laura Redondo-Flórez, Ana Isabel Beltrán-Velasco, Alexandra Martín-Rodríguez, Ismael Martínez-Guardado, Eduardo Navarro-Jiménez, Carmen Cecilia Laborde-Cárdenas, José Francisco Tornero-Aguilera

**Affiliations:** 1Faculty of Sports Sciences, Universidad Europea de Madrid, Tajo Street, s/n, 28670 Madrid, Spain; vctxente@yahoo.es (V.J.C.-S.); josefrancisco.tornero@universidadeuropea.es (J.F.T.-A.); 2Department of Health Sciences, Faculty of Biomedical and Health Sciences, Universidad Europea de Madrid, C/Tajo s/n, 28670 Madrid, Spain; lauraredondo_1@hotmail.com; 3Department of Psychology, Faculty of Life and Natural Sciences, University of Nebrija, C/del Hostal, 28248 Madrid, Spain; abeltranv@nebrija.es; 4BRABE Group, Department of Psychology, Faculty of Life and Natural Sciences, University of Nebrija, C/del Hostal, 28248 Madrid, Spain; imartinezgu@nebrija.es; 5Facultad de Ciencias de la Salud, Universidad Simón Bolívar, Barranquilla 080002, Colombia; 6Vicerrectoría de Investigación e Innovación, Universidad Simón Bolívar, Barranquilla 080005, Colombia; cacelaca6@gmail.com

**Keywords:** adipokines, inflammation, cardiovascular disease, atherosclerosis disease, mental disease, eating behavior, metabolic disease, cancer, microbiota, nutrition, physical activity

## Abstract

Adipokines are cell-signaling proteins secreted by adipose tissue that has been related to a low-grade state of inflammation and different pathologies. The present review aims to analyze the role of adipokines in health and disease in order to understand the important functions and effects of these cytokines. For this aim, the present review delves into the type of adipocytes and the cytokines produced, as well as their functions; the relations of adipokines in inflammation and different diseases such as cardiovascular, atherosclerosis, mental diseases, metabolic disorders, cancer, and eating behaviors; and finally, the role of microbiota, nutrition, and physical activity in adipokines is discussed. This information would allow for a better understanding of these important cytokines and their effects on body organisms.

## 1. Introduction

Adipokines are bioactive molecules secreted by adipose tissue that have various effects on health and disease. They play important roles in regulating metabolism, inflammation, immunity, cardiovascular function, and cancer. However, adipokine dysregulation can contribute to obesity-related disorders [1]. As there is a significant need to comprehend the mechanisms and interactions of adipokines, we deemed this review pertinent to the development of novel therapeutic strategies for these diseases. The review will therefore examine the current understanding of adipokines and their implications for health and disease. In addition to discussing the contradictory results regarding their parameters, we will also specify the severity of dysregulation in sections devoted to further explication in various regions of our organism.

In this line, adipose tissue is not only a passive storage organ for energy but also an active endocrine organ that secretes various molecules called adipokines. Adipokines are cytokines that regulate inflammation, metabolism, appetite, cardiovascular function, immunity, and other physiological processes. Adipokines include leptin, adiponectin, resistin, and many others. The type and amount of adipokines produced by adipose tissue depend on the type of adipocytes (white or brown), their size, number, location, and interaction with other cells. Adipocytes can be classified into two main types: white adipocytes and brown adipocytes [2]. White adipocytes store excess energy as triglycerides and secrete various cytokines such as leptin, adipsin, adiponectin, omentin, tumor necrosis factor-alpha (TNF-α), interleukin-6 (IL-6), monocyte chemoattractant protein-1 (MCP-1), plasminogen activator inhibitor-1 (PAI-1), resistin, visfatin, and retinol-binding protein 4 (RBP4) [3]. On the other hand, brown adipocytes store energy as small lipid droplets and secrete cytokines such as fibroblast growth factor 21 (FGF21), bone morphogenetic protein 7 (BMP-7), vascular endothelial growth factor A (VEGF-A), irisin, neuregulin 4 (NRG4), nesfatin-1, meteorin-like protein (METRNL), chemerin, IL-6, interleukin-8 (IL-8), interleukin-10 (IL-10). These cytokines have beneficial effects on thermogenesis, energy expenditure, glucose homeostasis, lipid metabolism, insulin sensitivity, angiogenesis, and anti-inflammation [4].

By modulating immune cell production of pro- and anti-inflammatory cytokines, they can modulate inflammatory responses. By modulating blood pressure, vascular tone, endothelial function, lipid metabolism, and thrombosis, they can also influence cardiovascular function. In addition, they can influence the progression of atherosclerosis by affecting plaque stability, smooth-muscle cell proliferation, macrophage infiltration, and foam-cell formation [5]. Furthermore, adipokines are also involved in mental disorders such as depression, anxiety, schizophrenia, bipolar disorder, and eating disorders. They can affect brain function by crossing the blood–brain barrier or activating receptors on neurons or glial cells. They can also modulate mood, cognition, stress response, reward system, and eating behavior through various neurotransmitters such as serotonin, dopamine, norepinephrine, and glutamate [6]. In this line, leptin, one of the adipokines, is known to play a role in regulating appetite and energy expenditure. It also has effects on mood and behavior, and studies have found that people with depression often have lower levels of circulating leptin [7]. Animal studies have shown that leptin can have an antidepressant-like effect, and some clinical trials have suggested that leptin administration may improve symptoms of depression in humans [8].

Therefore, alterations in the production of adipokines of an inflammatory nature mostly explain the appearance and epidemiology of Western diseases, including cancer and metabolic disorders [9]. Among these common diseases are diabetes, hypertension, atherosclerosis, fatty-liver disease, dementia, obstructive sleep apnea, and different types of cancer [1]. Concretely, these adipokines result in an alteration of glucose homeostasis, insulin sensitivity, lipolysis, and fatty-acid oxidation by acting on various tissues such as the liver, muscle, pancreas, and adipose tissue itself [10]. They can also contribute to other obesity-related complications such as nonalcoholic fatty liver disease (NAFLD), polycystic ovary syndrome (PCOS), and sleep-apnea syndrome (SAS) [11]. Related to cancer, adipokines can modulate tumor growth, angiogenesis, metastasis, apoptosis, and drug resistance by interacting with various signaling pathways such as NF-κB, JAK/STAT, MAPK, PI3K/Akt, and Wnt/β-catenin [12].

After all, these alterations in the adipocyte, and therefore in the production of adipokines of inflammatory etiology, could be largely regulated by nonpharmacological aids and as simple as physical exercise and nutrition, both with a fundamental implication in the modulation of another cofactor involved in the production of inflammatory and anti-inflammatory adipokines such as the intestinal microbiota. Microbiota can affect adipokine production and secretion by modulating adipose tissue inflammation, metabolism, immunity, and permeability. Research has shown that the composition and activity of the gut microbiota can have an impact on the production and function of adipokines [13]. Studies have found that certain bacteria in the gut can stimulate the production of adiponectin, which is a hormone that helps to regulate glucose and fatty-acid metabolism [14]. Other studies have shown that dysbiosis (imbalanced gut microbiota) is associated with increased levels of proinflammatory adipokines such as resistin [15]. Conversely, adipokines can also have an impact on the gut microbiota. For example, studies have found that leptin can influence the growth of certain types of gut bacteria, while adiponectin has been shown to have anti-inflammatory effects in the gut [13]. Overall, the relationship between adipokines and the gut microbiota is complex and multifaceted, with many factors influencing the interactions between these two systems.

Regarding nutrition, it can have a significant impact on the production and function of adipokines. For example, a diet high in saturated fats and simple sugars has been shown to increase the production of proinflammatory adipokines, such as resistin and IL-6, while a diet high in fruits, vegetables, and fiber has been associated with increased levels of anti-inflammatory adipokines, such as adiponectin [16]. Furthermore, certain nutrients and dietary components have been shown to directly influence the production and function of adipokines. For example, omega-3 fatty acids, found in fatty fish and certain plant-based sources, have been shown to increase the production of anti-inflammatory adipokines such as adiponectin, and decrease the production of proinflammatory adipokines, such as IL-6 [17]. Similarly, polyphenols, which are found in many plant-based foods such as fruits, vegetables, and tea, have been shown to have anti-inflammatory effects and may help to regulate the production of adipokines [18]. In addition to influencing the production and function of adipokines, nutrition can also have an impact on the gut microbiota, which in turn can affect adipokine production. For example, a diet high in fiber and prebiotic foods can promote the growth of beneficial gut bacteria, which have been shown to increase the production of adiponectin and decrease the production of proinflammatory adipokines [13]. Finally, physical exercise has shown that it can increase adiponectin levels in the body, which may contribute to the improved insulin sensitivity and glucose metabolism seen in active individuals [19]. Also, it is associated with increased levels of another adipokine called leptin, which regulates appetite and energy balance. Chronic overproduction of leptin can lead to leptin resistance, a condition where the body becomes less responsive to the appetite-regulating effects of the hormone [19].

Therefore, in order to analyze the role of adipokines in health and disease the present research has been conducted. Due to this, it focuses on the pathophysiological role of adipokines in disease and health, as well as in the study of nonpharmacological contributing factors such as exercise and nutrition.

## 2. Methods

The protocol used consisted of a literature search, using primary sources, such as scientific articles, and secondary, such as bibliographic indexes, databases, and web pages. We used PubMed, Embase, SciELO, Science Direct Scopus, and Web of Science, employing MeSH-compliant keywords, including adipokines, adipose tissue, eating behaviors, inflammation, metabolic disease, cancer, gut microbiota, atherosclerosis, mental diseases, nutrition, and physical activity. We used articles and information published from 2001 to 2023, although previous studies from the 2000s and earlier were included to explain some information in several points of the review. From a total of 22,033 based on the initial inquiry, the following exclusion criteria were applied: (i). studies with inappropriate or not relevant topics, being not pertinent to the main focus of the review, and (ii). Ph.D. dissertations, conference proceedings, and unpublished studies. We included all the articles that met the scientific methodological standards and had implications with any of the subsections in which this article is distributed: obesity and adipose-tissue dysregulation, functions of adipokines, adipokines, and inflammation, adipokines and cardiovascular disease, adipokines and atherosclerosis, adipokines, and metabolic disease, adipokines and mental disease, adipokines and eating behaviors, adipokines and cancer, the role of microbiota in adipokines, the role of nutrition in adipokines, and the role of physical activity in adipokines. The information treatment was performed by the eight authors of the review, and each subtopic was decided by consensus. Articles were discussed by the authors and the final 322 papers were read to be considered relevant to the search criteria and appropriate for assessing our research objective.

## 3. Obesity and Adipose Tissue Dysregulation

Adipose tissue is composed of various types of adipocytes, which are specialized cells responsible for storing and releasing energy in the form of triglycerides [20]. Adipocytes also play a crucial role in the regulation of energy homeostasis by secreting various cytokines, including adiponectin, leptin, IL-6, and TNF-α [21]. The differential expression of cytokines by different types of adipocytes contributes to the development of metabolic disorders, including obesity, insulin resistance, and type two diabetes.

### 3.1. Obesity Pandemic

At the conclusion of the 20th century, obesity emerged as a global health concern. A pandemic of obesity is currently recognized [22]. The availability of highly processed foods that are very simple to handle or do not require handling and, most importantly, are very inexpensive contributes significantly to the persistent rise in obesity rates. The information presented by authors from around the globe is alarming [23]. When energy intake consistently surpasses energy expenditure, obesity follows. Numerous variables, including genetic, epigenetic, hormonal, and lifestyle factors, are involved. Adipose tissue can grow via either hypertrophy (an increase in adipocyte size) or hyperplasia (an increase in adipocyte quantity as a result of the recruitment of new adipocytes) during the development of obesity [24]. When caloric intake is excessive, adipocytes first become hypertrophic and secrete adipokines that cause the recruitment of additional pre-adipocytes, which then differentiate into mature adipocytes as a protective measure against some of the negative metabolic effects of obesity [25]. In the following sections, these mechanisms as well as the differentiation between the various types of adipose tissues and the secretion of adipokines in each of them will be discussed.

### 3.2. White Adipocytes

White adipocytes, also known as white adipose tissue (WAT), are the most common type of adipocytes found in the human body. These cells are primarily responsible for storing excess energy in the form of triglycerides. White adipocytes produce several cytokines, including adiponectin, leptin, and IL-6 [21]. Adiponectin is an insulin-sensitizing cytokine that plays a crucial role in glucose homeostasis and lipid metabolism. Leptin, on the other hand, is a hormone that regulates energy balance by suppressing appetite and increasing energy expenditure. IL-6 is a proinflammatory cytokine that is involved in the immune response and plays a role in insulin resistance [26].

One of the key cytokines produced by white adipocytes is leptin. Leptin is an adipokine that is produced primarily by white adipose tissue and is involved in the regulation of appetite and energy balance. Leptin levels are positively correlated with body-fat mass, and leptin resistance is a hallmark of obesity and metabolic disorders such as type two diabetes [27]. White adipocytes also produce a number of proinflammatory cytokines, including IL-6 and TNF-α. These cytokines have been implicated in the development of insulin resistance, a key feature of type two diabetes and other metabolic disorders [28]. IL-6 is produced by white adipocytes in response to various stimuli, including exercise, and has been shown to have both pro- and anti-inflammatory effects [29]. TNF-α, on the other hand, is a potent proinflammatory cytokine that is produced by a variety of cells, including white adipocytes [30]. Recent studies have also highlighted the role of other cytokines produced by white adipocytes, including resistin and adiponectin. Resistin is a cytokine that is produced by white adipocytes and has been implicated in the development of insulin resistance and other metabolic disorders. Adiponectin, on the other hand, is an anti-inflammatory cytokine that is produced by adipocytes and has been shown to have beneficial effects on glucose and lipid metabolism [31].

Another cytokine produced by white adipocytes is visfatin, which has been shown to be involved in insulin sensitivity and glucose homeostasis [32]. Visfatin is also known as nicotinamide phosphoribosyltransferase (NAMPT), which plays a role in the biosynthesis of nicotinamide adenine dinucleotide (NAD+), an important cofactor in energy metabolism [33]. However, the role of visfatin in metabolic disorders is still controversial and further research is needed to elucidate its exact functions. White adipocytes also produce chemokines such as MCP-1, which plays a role in the recruitment of macrophages to adipose tissue and the development of adipose tissue inflammation. Adipose tissue inflammation is thought to be a key factor in the development of insulin resistance and other metabolic disorders [33].

### 3.3. Brown Adipocytes

Brown adipocytes, also known as brown adipose tissue (BAT), are found in small amounts in the human body, primarily in newborns and hibernating animals. These cells are specialized for thermogenesis and are responsible for generating heat by burning fat. Brown adipocytes produce various cytokines, including FGF21, IL-6, and TNF-α. FGF21 is a hormone that regulates glucose and lipid metabolism, while IL-6 and TNF-α are proinflammatory cytokines that play a role in insulin resistance [34]. One of the most well-known cytokines produced by brown adipocytes is irisin, which has been shown to increase energy expenditure and improve glucose homeostasis. Irisin is derived from the cleavage of fibronectin type III domain-containing protein five (FNDC5) and is released into circulation by brown adipocytes in response to cold exposure or exercise [35]. Another cytokine produced by brown adipocytes is FGF21, which has been shown to improve insulin sensitivity and lipid metabolism. FGF21 is secreted by brown adipocytes in response to cold exposure and acts on multiple tissues to improve metabolic function [36].

Brown adipocytes also produce IL-6, which plays a role in thermogenesis and the regulation of energy expenditure. IL-6 has been shown to stimulate brown adipocyte differentiation and thermogenesis through the activation of the cAMP/PKA pathway [37]. In addition to cytokines, brown adipocytes also produce various other bioactive molecules, including adipokines such as adiponectin, which has been shown to improve insulin sensitivity and lipid metabolism. Adiponectin is also known to regulate inflammation and play a role in the development of metabolic disorders [38]. Brown adipocytes also produce extracellular vesicles (EVs), which have been shown to play a role in the regulation of energy metabolism and intercellular communication. Brown adipocyte-derived EVs have been shown to contain various bioactive molecules, including microRNAs (miRNAs), which have been implicated in the regulation of brown adipocyte function and thermogenesis [39].

### 3.4. Beige Adipocytes

Beige adipocytes, also known as brite (brown-in-white) adipocytes, are a subtype of white adipocytes that have brown-like characteristics. These cells can be induced to differentiate from white adipocytes in response to cold exposure or pharmacological agents such as β-adrenergic agonists. Beige adipocytes produce cytokines such as IL-6, TNF-α, and FGF21. These cytokines play a crucial role in the regulation of glucose and lipid metabolism and energy homeostasis [40].

Beige adipocytes, also known as brite (brown-in-white) adipocytes, are a distinct type of adipocyte that exhibit characteristics of both white and brown adipocytes. Beige adipocytes are induced by various stimuli such as cold exposure, exercise, and β-adrenergic agonists, and have been shown to play a role in the regulation of energy expenditure and metabolic function [41]. Beige adipocytes produce several cytokines that regulate energy metabolism, including IL-6, which has been shown to promote the browning of white adipocytes and enhance thermogenesis. IL-6 also plays a role in the regulation of glucose metabolism and insulin sensitivity [42]. Another cytokine produced by beige adipocytes is tumor necrosis factor-alpha (TNF-α), which has been shown to regulate the browning of white adipocytes and improve insulin sensitivity. TNF-α also plays a role in the regulation of inflammation and immune function [41].

Beige adipocytes also produce several other bioactive molecules that play a role in the regulation of energy metabolism, including fibroblast growth factor 21 (FGF21), which has been shown to improve glucose metabolism and insulin sensitivity. FGF21 is induced by various stimuli such as cold exposure, exercise, and high-fat diets, and acts on multiple tissues to improve metabolic function [36].In addition to cytokines, beige adipocytes also produce extracellular vesicles (EVs), which have been shown to play a role in intercellular communication and the regulation of energy metabolism. Beige adipocyte-derived EVs have been shown to contain various bioactive molecules, including miRNAs, which have been implicated in the regulation of beige adipocyte function and thermogenesis [43].

Recent studies have shown that manipulating the expression of cytokines produced by adipocytes can affect metabolic outcomes. For example, overexpression of IL-6 in white adipose tissue has been shown to promote insulin resistance and glucose intolerance in mice [44]. Similarly, the deletion of TNF-α from white adipose tissue has been shown to improve insulin sensitivity and glucose tolerance in mice [21]. These findings suggest that targeting cytokine production with different types of adipocytes could have therapeutic potential for the treatment of metabolic disorders. Furthermore, recent studies have also demonstrated the potential of inducing the browning of white adipocytes as a therapeutic strategy for metabolic diseases. Activation of beige adipocytes in white adipose tissue has been shown to increase energy expenditure and improve glucose and lipid metabolism [20]. This approach may involve targeting various signaling pathways involved in adipocyte differentiation, including the peroxisome proliferator-activated receptor-γ (PPAR-γ) and the β3-adrenergic receptor (β3-AR) pathways.

## 4. Adipokines and Inflammation

Adipose tissue was long thought to be merely an energy storage organ. However, research was done recently, and initiatives to comprehend obesity have produced new knowledge showing that it is not only an active organ but also an inflammatory [45]. Among the endocrine factors elaborated by adipose tissue are the adipokines, pro- and anti-inflammatory molecules created in response to alterations in adipocyte triacylglycerol accumulation but also in response to both local and systemic inflammation. These alterations could affect long-term energy storage and have a major effect on reproductive function, blood pressure management, energy balance, the immune system, and a great number of other physiologic functions [2]. Numerous variables, such as the host’s nutritional and metabolic state, the presence of this infection or inflammation, oxidative stress, smoking status, age, and sex, control the ratio of pro- and anti-inflammatory adipokines [2]. Due to the fact that the majority of adipokines are elevated in obesity and contribute to the “low-grade inflammatory state” associated with obesity, adipokines are currently regarded as significant players in inflammation and immunity [46].

Even while we are emphasizing the crucial part adipokines play in controlling the inflammatory response, they are not the only players in this scenario. In essence, adipose tissue is made up of mature adipocytes, preadipocytes, mesenchymal cells, and stromal vascular fraction (SVF) cells, which also include fibroblasts, vascular endothelium, smooth muscle cells, and other leukocyte subsets. It is interesting to note that adipose tissue contains practically all immune cells, including resident macrophages, mast cells, monocytes, dendritic cells, natural killer cells, B-cells, T-cells, neutrophils, and eosinophils [47,48,49]. Nonetheless, the activation state, differentiation, and proliferation of these immune cells in adipose tissue are dramatically impacted by anti- and proinflammatory cytokines, lipid mediators, and adipokines produced within local fat pads and the bloodstream [50]. Since the dysregulation of substances generated by adipose tissue has metabolic effects that contribute to the pathogenesis of society disorders [51], its occurrence must be specified. Similar clinical problems, such as hypertriglyceridemia, insulin resistance, and fatty liver, result in diabetes, hypertension, polycystic ovarian syndrome (PCOS), coronary artery disease (CAD), and cancer [52,53,54]. We consider that it is essential to describe which pathway each component follows in order to clarify this process by differentiating proinflammatory and anti-inflammatory adipokines.

### 4.1. Proinflammatory Adipokines

The discovery of leptin, the first adipokine, was a significant step forward in the understanding of adipose tissue as an endocrine organ [55]. It was first described as a satiety hormone due to regulates body weight by inhibiting food intake and stimulating energy expenditure via a feedback loop between adipocytes and the hypothalamus. However, evidence revealed that leptin functions as both a hormone and a cytokine. As a hormone, it regulates many endocrine activities including bone metabolism, in addition to its primary function of influencing energy homeostasis via thermoregulation-related mechanisms [45,56]. Moreover, it stimulates inflammatory reactions as a cytokine. It follows that elevated levels of circulating leptin in obese people considerably contribute to the low-grade inflammatory state that makes these individuals more prone to cardiovascular disease, type II diabetes, or degenerative illness [27,57]. In general, leptin has proinflammatory properties [14,15] and it can directly enhance the generation of many proinflammatory cytokines, such as IL-6, interleukin 12 (IL-12), interleukin18 (IL-18), and TNFα, the chemokines IL-8 and CCL2/MCP-1, and the lipid intermediaries PGE, cysteinyl leukotrienes (cysLTs), and leukotriene B4 (LTB4) in peripheral blood monocytes [27,58]. Conversely, reduced levels of leptin have been linked to an increased risk of infection and reduced cell-mediated immunity [59,60]

In mouse models of diet- and genetically-induced obesity, resistin levels in the blood are elevated relative to other proinflammatory adipokines, suggesting that high resistin levels are related to obesity and associated metabolic dysfunctions [61]. Similarly, CAP1, which was discovered as a resistin receptor just recently, mediates the proinflammatory actions of resistin. It promotes cAMP-mediated PKA activation and NF-B-related inflammatory cytokine production in human monocytes [62]. Several other poorly researched proinflammatory adipokines, such as chemerin, RBP4, and lipocalin two, have been linked to the increase of inflammation associated with obesity (LCN2). Chemerin is an adipocyte-derived chemoattractant for monocytes and dendritic cells that are produced by mature adipocytes [63]. RBP4 may promote inflammation by stimulating antigen-presenting cells in adipose tissue, which promotes TH1 cell polarisation [64]. LCN2 is generated by adipocytes and is activated by inflammatory stimuli in adipose tissue [65].

### 4.2. Antiinflamatory Adipokines

Regarding other adipokines with anti-inflammatory effects, such as CTRP, omentin, and secreted frizzled-related protein five (SFRP5), relatively little is known [66,67]. Adiponectin is the most well-known and abundant adipokine detected in human serum, with average concentrations in the range of g/mL. Adipocytes from white adipose tissue and brown adipose tissue express the adipokine adiponectin, which is antisteatotic, anti-inflammatory, and antifibrotic [68]. Recent research has shown that adiponectin appears to have both proinflammatory and anti-inflammatory effects. It acts on insulin sensitivity, and its plasma lowering is associated with insulin resistance and glucose intolerance, hence increasing the risk of progressive liver injury. However, it contributes significantly to anti-inflammatory action, thermogenesis, lipolysis, and oxidation of fatty acids in skeletal muscle and the liver [69]. Thus, A few anti-inflammatory adipokines that are downregulated in the states of obesity appear to guard against the emergence of obese comorbidities [70].

In contrast to other adipokines, which are most abundant in visceral and subcutaneous adipose tissue (SCAT), it is produced primarily by bone-marrow adipose tissue (BMAT). Its effects are mediated by the AdipoR1 and AdipoR2 receptors, which in immune cells and tissues activate AMPK. Evidence has demonstrated that it inhibits vascular inflammation, which, for instance, may protect against aortic aneurysms [71]. Nevertheless, low blood adiponectin levels are associated with chronic inflammation of metabolic diseases such as type two diabetes, obesity, and atherosclerosis [72].

Additionally, by blocking TLR4 activation, C1q/TNF-related protein three (CTRP3) has been demonstrated to lower cytokine production in human monocytes and adipocytes challenged with lipopolysaccharide and free fatty acids. CTRP13 also reduces inflammation in lipid-loaded hepatocytes and increases insulin sensitivity [73]. Omentin is a new adipokine that suppresses TNF α-induced COX2 expression in endothelial cells and stimulates endothelial nitric oxide synthase [74]. The anti-inflammatory effects of SFRP5 in adipose tissue and macrophages are mediated by the inhibition of noncanonical Wnt5a/JNK signaling, which in turn reduces macrophage TNF α, interleukin one (IL-1), and CCL–MCP1 synthesis [75]. A greater comprehension of the biology of these novel anti-inflammatory adipokines and their role in combating the effects of proinflammatory mediators and adipokines is required.

To sum up, the proinflammatory mediators known as adipokines, in which TNF α, monocyte chemoattractant protein MCP1, and IL-6, speed up the development of illness. In contrast, a few anti-inflammatory adipokines that are downregulated in states of obesity appear to offer protection against the emergence of obese comorbidities [66]. By differentiating between lean tissue and fat tissue, we can identify several distinct health effects (Figure 1). Concretely, the dysregulation of the production of adipokines seen in obesity is linked to the pathogenesis of various disease processes, especially metabolic and cardiovascular diseases [76] (Figure 1). Particularly, the increased synthesis of anti-inflammatory adipokines could be useful as a potential treatment for a variety of obesity-related problems [66].

## 5. Adipokines and Cardiovascular Disease

Cardiovascular disease (CVD) has been largely described by its great impact on global mortality. It was also pointed out as an important key factor related to a large number of comorbidities, compromising overall health status [77]. The recent literature proposed how a special kind of cytokines may have a great impact on CVD due to their capability of modulating different cellular processes.

Adiponectin is one of the amplest adipokines studied due to its apparently negative association with diabetes mellitus, obesity, and CVD [78,79]. Regarding CVD, decreased adiponectin levels have been related to a raised risk of high blood pressure as well as to cardiomyopathy in hyperglycemic patients [80,81]. Thus, hyperadiponectinemia has been elucidated as a protective cardiovascular factor, since it contributes to increasing nitric oxide (NO) release from the endothelium and, consequently, may decrease vascular dysfunction, as well as it may reduce adhesion-molecules expression [82]. Nevertheless, recent studies suggested the controversial role of adiponectin as a protective agent in CVD, since increased adiponectin levels were involved in the increased risk of coronary-artery disease [83], atrial fibrillation [84] and heart failure [85], as well as raised levels were not linked to higher stroke risk [86,87]. Consequently, we consider further studies are needed in order to clarify if adiponectin may constitute a risk factor for cardiovascular events or if it may be considered a risk marker found in different CVD.

Leptin is another of the most studied adipokines due to its relationship to obesity and cardiovascular events [88]. Leptin levels are positively associated with adipose tissue presence, since the higher the fat mass, the greater leptinemia could be found in patients [89]. Leptin is involved in several types of cellular dysfunction, such as proinflammatory processes, oxidative stress, angiogenesis enhancement, atherogenesis and thrombotic events, and endothelial dysfunction improvement as well as arterial tightness and atherosclerotic plaque development [58,90]. Due to its effects on the cardiovascular system, several studies pointed out how hyperleptinemia may be associated with myocardial infarction in addition to raised leptin levels [91], which have been associated with coronary heart disease [92,93], including a higher risk of cardiac death, nonfatal myocardial infarction, unstable angina, cerebrovascular accident, and heart failure [94]. Regarding endothelium status, leptin has been pointed out as an important key factor in platelet activation processes in coronary heart-disease patients, as well as by its ability to enhance tissue factor expression and adhesion-molecules development, compromising endothelium functioning [95,96]. Nevertheless, several controversial results have been elucidated, since a recent meta-analysis suggested that there is no association between raised leptin levels and coronary heart-disease risk [97]. Similar findings were reported in the recent literature, in which it was proposed that hyperleptinemia and increased body mass index could be considered as predictors of a better prognosis in coronary heart-disease patients [98]. Furthermore, raised leptin levels were associated with a lower incidence of adverse events, morbidity, and mortality in coronary heart-disease patients [99].

Chemerin is an adipokine known for its ability to participate in adipogenesis and angiogenesis, as well as by its participation in inflammatory processes, due to the fact that it plays an important role as a chemoattractant. Increased chemerin values have been related to different cardiovascular happenings, such as an increased risk and severity of coronary arterial disease [100], probably explained by its activity promoting endothelial dysfunction and raising arterial rigidity [101]. Additionally, chemerin’s effect in CVD also could be explained through its role as a chemoattractant, as the previous literature pointed out, wherein raised chemerin levels were found along with other inflammatory cytokines, including IL-6, C-reactive protein, and TNF-α [102]. These findings suggest chemerin could be responsible for enhancing inflammation in the endothelium as well as perhaps promoting angiogenesis.

Resistin is an adipokin that could be released from epicardial adipose tissue and it has been linked to insulin resistance as well as to inflammatory processes [103]. Thus, it plays an important role in proinflammatory cytokines secretion, including IL-1β, IL-6, IL-8, IL-12, TNF-α, and MCP-1, as well as in nuclear factor-κB (NF-κB) activation [104]. Consequently, it triggers systemic tissue inflammation, oxidative stress, and endothelial dysfunction [105], which may compromise the vascular system. Regarding CVD, elevated resistin values have been associated with ventricular dysfunction and atrial fibrillation risk, due to resistin’s capability of triggering myocardial fibrosis [103]. In relation to other CVDs, the opposite results were described in the previous literature, wherein resistin was highlighted as an independent risk factor for coronary arterial disease, myocardial infarction, and heart failure, whereas it is linked to atrial fibrillation [106].

RBP4 is an adipokine that is secreted by the liver and adipose tissue. This adipokine showed proinflammatory properties, probably due to its activity through Toll-like receptor four, leading to inflammation and hypertrophic effects, which could compromise CVD pathogenesis [107]. Then, it has been related to heart failure, since raised levels were found in advanced heart failure patients as well as being elucidated as a worse prognostic indicator in elderly patients with chronic heart failure [108,109]. Furthermore, recent research proposed how RPB4 may have prohypertrophic effects since increased RPB4 levels were found in patients who showed an elevated left-ventricular mass index and left-atrial end-systolic dimension, which could trigger heart failure [110].

CVD has a great impact on overall health status, as we mentioned above. In this line, recent research developed in 2022 focused its efforts on genetic prediction through a systematic exploration of genome-wide association studies for four circulating adipokines. Interesting findings have been obtained as a result of this study, where elevated genetically predicted chemerin values may be related to an increase in coronary-artery disease risk. Additionally, greater genetically predicted resistin levels are significantly related to a raised atrial-fibrillation risk. Finally, increased genetically predicted retinol-binding protein four levels could be associated with a rise in heart-failure risk [111]. These findings support the importance of adequate metabolic-balance maintenance, where adipokines production by adipose tissue is controlled since many studies have linked these substances to an increased risk of CVD.

## 6. Adipokines and Atherosclerosis

Atherosclerosis has been extensively related to cardiovascular and metabolic events in literature, being the most important risk factor for CVD and stroke [112]. Thus, atherosclerosis has been linked to coronary artery disease due to the important role of adipokines in both pathways [113,114]. It is important to consider that atherosclerosis may enhance cardiovascular disease, and consequently, compromise health status and quality of life.

In this line, adiponectin has shown a protective effect in cases of fibrotic and inflammatory events [115,116], which has been linked to a beneficial result against atherosclerosis [117]. Thus, it may be explained due to its capability to promote fatty-acids catabolism [118]. Additionally, it also may be explained by the reduction in triglyceride aggregation triggered by adiponectin’s enhancement in fatty-acid oxidation in skeletal muscle [119]. Furthermore, adiponectin levels were found to decrease in obese subjects, as these lower adiponectin levels could be considered a risk factor in metabolic syndrome patients [120].

As we mentioned above, leptin is an adipokine that may enhance atherogenesis, being that levels were found raised in patients who presented coronary atherosclerosis [121], as well as both ischemic and hemorrhagic ictus [122]. Furthermore, the previous literature highlighted leptin activities, including reactive oxygen species (ROS) generation triggering oxidative stress, enhancement of endothelial cells growth, angiogenesis regulation, and also modulating adhesion molecules and tissue-factor expression [123]. Then, according to leptin’s effects in vessels and its association with proinflammatory events, this adipokine has been mostly linked to endothelial dysfunction and the related vascular pathways [124].

Resistin levels were found to increase in obese patients, a fact which also could be related to metabolic syndrome. Then, it could be explained through resistin’s effect in inflammation-induced insulin resistance, probably due to its activity over inflammatory cytokines, such as IL6, IL-12, TNF-α, and (NF-κB). In this line, considering this inflammatory influence is due to cytokines, resistin also may play an important role in endothelial functioning, promoting the initiation and progression of atherosclerosis [55,125]. Additionally, it has been proposed by previous literature how resistin may contribute to endothelin-1 expression in endothelial cells, triggering endothelial dysfunction, a fact that may compromise atherosclerotic vessels. Furthermore, resistin has been elucidated as responsible for the raised expression of vascular cell adhesion protein one (VCAM-1) and MCP-1. Both proteins, VCAM-1 and MCP-1 have been related to early atherosclerotic injury development as well as to smooth muscle cell production, suggesting the early damage in the vascular system in hyperglycemic patients [126]. Moreover, patients who suffered acute coronary syndrome showed increased resistin levels in the early onset of the episode, suggesting that resistin may have a negative impact on plaque stability [127]. Finally, as is the case with leptin, resistin was also related to ROS generation, endothelial-cell progress, angiogenesis improvement, and an increase in adhesion-molecules expression [128].

Chemerin also has been associated with endothelium dysfunctioning and cardiovascular disease, due to its proinflammatory activity [129]. Additionally, it also shares the ability of leptin and resistin in terms of promoting ROS production, endothelial cell generation, platelet adhesion, and angiogenesis [130]. Thus, it was reported in recent research that patients who suffered coronary artery disease showed increased chemerin levels, suggesting that coronary artery disease risk may be related to raised chemerin values [83]. Furthermore, chemerin plays an important role in enhancing fat cell maturation, triglycerides production, and glucose transport, as well as by its capability to inhibit lipolysis [131]. As a consequence, considering chemerin’s effect as a proinflammatory molecule added to the fact that it may compromise endothelium status, it could be said that chemerin might have a negative impact on atheroma plaque grade and stability. Similar results were found in recent literature, where chemerine is proposed as a higher atherosclerotic risk marker [132].

Finally, RPB4 also has been lately related to atherosclerosis in mice models, since raised RPB4 levels are linked with atherosclerosis in diabetic rats. Then, it has been recently indicated by previous authors how RPB4 may have a key role in the JAK2–STAT3 kinases signaling route. These findings suggest that RBP4 may be implied in JAK2 and STAT3 phosphorylation, leading to atherosclerosis creation through the JAK2–STAT3 pathway [133]. Furthermore, RPB4 promotes vessel inflammation which is probably associated with other inflammatory cytokines, such as IL6, TNF-α, VCAM-1, and intercellular cell adhesion molecule-1 (ICAM-1) in endothelial cells. Additionally, it has been suggested that RPB4 may also enhance atherogenesis by prompting macrophage-derived foam cell formation [134]. Thus, it could be said that RPB4 is a recently studied molecule compared to other ones, which also yields information about the risk of atheroma plaque production and its implication in CVD.

## 7. Adipokines and Mental Disease

It is known that adipokines constitute a very active metabolic complex that is directly related to organ homeostasis. This complex structure can modify morphology and metabolic functions to adapt to the physiological characteristics of the organism.

Specifically, leptin plays a fundamental role in energy homeostasis and is produced by adipose tissue. This adipokine plays an essential role in appetite regulation through the arcuate nucleus of the hypothalamus in the central nervous system [135]. This regulation allows metabolic expenditure to be increased and the sensation of hunger to be reduced. It is also involved in the regulation of insulin sensitivity, a factor that is part of the so-called metabolic syndrome [136]. In relation to the metabolic syndrome, other adipokines also act, such as resistin, which facilitates insulin resistance, and adiponectin, which regulates insulin sensitivity and maintains anti-inflammatory effects associated with the modulation of endothelial damage [137]. Metabolic syndrome is considered a risk factor in cardiovascular pathologies and diabetes, and although the associated symptoms are not entirely clear, it is associated with different metabolic alterations. This pathology is also present in other mental disorders and its incidence is higher in these patients than in the general population [138].

In reference to psychotic disorders, a high prevalence of comorbidity with metabolic syndrome has been found [139]. It is not known with certainty whether the endocrine imbalance appears because of the medication applied in these pathologies (antipsychotics), whether it is the mental pathology itself that triggers a metabolic alteration, or whether patients with mental disorders are really more prone to maintain certain nutritional habits that favor the appearance of the metabolic syndrome [140].

However, the presence of primary pathophysiological components has been associated with mental pathologies [135]. Patients with these pathologies tend to have unhealthy lifestyles, which may predispose them to generate this syndrome. Studies in this line show that they may also present some molecular factors that make them vulnerable to these lifestyles and their impact on the organism, such as high levels of anxiety or depressive symptomatology, which facilitates alterations in sensitivity to glucocorticoids, which destabilizes the regulation of the hypothalamic–pituitary–adrenal axis. It is known that high adrenocortical stimulation is associated with lipolytic pathway activity. This hyperactivation increases free triglycerides and fatty acids, leading to the presence of dyslipidemia [141].

The microbiota is also important in mental pathologies, as it regulates the production and secretion of signal molecules that are associated with neuronal plasticity and brain development. Inflammation maintained over time is a sign of alterations that are present in both metabolic syndrome and psychopathologies [142]. Other studies have been able to demonstrate the relationship between cardiometabolic alterations in people with severe mental illness and some pleiotropic genes [143]. In addition, another study has shown that leptin and other serotonergic 2C receptor genes may be involved in the metabolic syndrome developed in patients with severe mental illnesses such as schizophrenia [144].

The metabolic factor associated with mental illness is especially significant in major depressive disorder (MDD). This disorder has a high rate of comorbidity with cardiovascular disease and the risk of death it presents [145]. This is due to the body’s response to an elevated level of stress and its relationship to the secretion of hormones such as vasopressin and corticotropin. Corticotropin projection neurons act directly on the spinal cord and brainstem, causing the activation of neurons of the autonomic nervous system, and sympathetic branch, and inhibiting the action of parasympathetic neurons through the stimulation of α2 receptors. This activation will allow the release of corticotropin in the hypothalamus [145].

In addition, this increase in corticotropin will act on the immune system, specifically on macrophages, and stimulate the production of proinflammatory cytokines. Patients with MDD have higher levels of proinflammatory cytokines than the general population, and more receptors in the cerebrospinal fluid [146]. Other studies related to the presence of leptin in patients with MDD indicate that the hormone may have antidepressant effects. However, when there is a deregulation of this adipokine, its effects are nullified or diminished by the action of a proinflammatory diet and the presence of hyperleptinemia [147].

Metabolic syndrome in mental patients has also been associated with the effect of an increase in ROS, very small molecules with high reactivity that can modulate proinflammatory genes, which are related to oxidative stress [148]. The effect of reactive oxygen species is associated with lesions in pancreatic cells and the reduction of enzymes with an antioxidant effect. Therefore, alterations in these cells facilitate a decrease in insulin secretion [149]. All these elements that have been observed in patients with mental illness should be considered as mechanisms that are established in the pathophysiology of these individuals and that have a direct impact on the maintenance of the psychopathology presented. Endocrine action must be combined with the psychological and psychiatric perspective for the patient’s recovery since the repercussions of metabolic activity are directly related to neuronal activity and its connections with the different organs. 

## 8. Adipokines and Eating Behaviors

In recent years, the relevant role of the impact of some biological factors that are involved in eating behaviors and in the disorders that are triggered due to the application of these behaviors has been considered. In addition to genetics and other vulnerability elements such as gender and age, the study of cytokines and their alterations has been related to the presence of certain eating disorders [142]. One of the most important effects of adipokines is to maintain metabolic homeostasis through sensitization or resistance to insulin. Therefore, they all have a similar effect to insulin, reducing blood glucose levels [150].

Interleukin 15 (IL-15) is a cytokine that has a structure similar to IL-2 and is linked by a complex system composed of the beta chain of the CD122 receptor and the gamma-C chain. This cytokine is secreted by nuclear phagocytes and modulates the activity of T cells and NK cells, which attack viruses present in the body [151]. VCAM-1 is expressed by the arterial endothelium, which is a monolayer that separates blood from tissues and regulates systemic flow by modifications in vascular tone [152]. VCAM-1 is involved in the attraction of cells whose function is atherosclerotic. High values of these cytokines are associated with patients diagnosed with anorexia and other eating disorders [153].

Interleukin 6 (IL-6) is a cytokine that plays an essential role in the immune system and has several functions [154]. It is involved in the intervention against infections, but it also has a fundamental role in the activation of fever and in the production of proteins in different organs such as the liver. They also modulate immune function and coagulation. Among others, they regulate TH17 and regulatory T lymphocytes and regulate the organic inflammatory response. Alterations in this cytokine are associated with autoimmune and inflammatory diseases. The appearance of elevated IL-6 levels can be observed in patients with eating disorders due to their inflammatory impact on the body [155].

The same occurs with TNF-α, tumor necrosis factor, and C-reactive protein, which have been extensively studied in recent years and the results in animals have shown that subclinical proinflammatory action and their relationship with the action of the immune system can lead to pathologies associated with obesity and metabolic alterations, and with insulin sensitization [156]. In people with obesity, the reduction of TNF-α accounts for a reduction in adipose tissue. This cellular dysfunction mediated by cytokines and proinflammatory mediators explains the eating habits of people who eventually suffer from an eating disorder [157].

The functioning of these adipokines, together with the action of environmental factors and other cognitive or psychological factors such as the presence of stress, favor the deterioration of the organism and the inflammation of the different organs [158]. Adipose tissue is formed by the presence of an increase in cytokines, especially white adipose tissue, and is regulated by different proteins that are involved in the regulation of blood pressure, coagulation, or vascularization [159].

BAT is related to temperature regulation and white adipose tissue to cytokine secretion and fat storage [160]. Both tissues are very heterogeneous in their functions, though their dysregulation is associated with inappropriate eating behaviors, obesity, and diseases associated with a chronic inflammatory response, which facilitates abnormal cytokine production and signaling activity in inflammatory response pathways.

At the base of the hypothalamus, where the integration of the signals that will transmit the adipokines takes place, is where we can see the damage that occurs [161]. When there are alterations in the paraventricular and ventromedial hypothalamic nuclei, this is when obesity and hyperphagia appear. On the other hand, damage in the lateral hypothalamus explains deregulated nutritional habits, anorexia, and general weight loss [162].

The hypothalamus is regulated by a cellular system composed of proopiomelanocortin and adipokines stimulate this cellular system and thereby reduce appetite [163]. For the person to reach satiety, the proopiomelanocortin system projects to the ventromedial nuclei which are called the satiety center. From this signaling circuit, certain neurotransmitters are released that will regulate the activation of the autonomic nervous system [164].

The homeostasis that occurs in the modulation channels of the central nervous system and the autonomic nervous system, in their physiological interaction, explains the regulation of complex metabolic processes [165]. These signals depend on the energy of the organism, and the nutrition that is maintained is essential to determine a correct caloric intake [166]. The neuronal populations of the arcuate nuclei have antagonistic effects in the modulation of eating behavior, since neuropeptides that have an anorexigenic action, among others, are going to be expressed here. On the other hand, the cortical centers of people are closely related to complex emotional and cognitive processes that stimulate food consumption ignoring the homeostatic needs of the organism [165].

It is possible to make an association between eating behaviors and nutritional patterns of individuals by understanding the functioning of the metabolic system and its relationship with the central nervous system and the autonomous nervous system through the hypothalamic–pituitary–adrenal axis and the implications of neural circuits in their communications between different regions of the body. This understanding should prove useful in the evaluations carried out on people with unhealthy or dysfunctional eating habits and be able to make appropriate and individualized interventions that consider these factors [167,168].

## 9. Adipokines and Metabolic Disease

Adipose tissue inflammation is initiated and sustained over time by aberrant adipocytes that secrete inflammatory adipokines and by the invasion of immune cells formed in the bone marrow that communicates via the production of cytokines and chemokines. Low-grade inflammation of adipose tissue has a detrimental effect on distant organ function and is believed to be the core cause of obesity complications [169]. Among them, visceral obesity plays an essential role in the development of metabolic syndrome [170]. It is diagnosed when at least three of the following five criteria are present: High waist–hip ratio, high blood pressure, high blood sugar, elevated triglycerides (TGs), and poor high-density lipoprotein (HDL) cholesterol [171]. Due to its link to rising diabetes incidence and a higher risk of cardiovascular events including heart disease and stroke, which have developed into significant public health concerns, metabolic syndrome is crucial. Certain adipokines can induce pathogenic diseases linked to obesity, lipid accumulation, and insulin resistance when they are dysregulated [172].

Thus, as a highly dynamic endocrine tissue, it serves as a central node in the interorgan crosstalk network via adipokines, which modulate angiogenesis, metabolism, and inflammation, among other effects [173]. Specifically, the liver is the metabolic nerve center for carbohydrate, lipid, and protein metabolism [174]. Adipose tissue and the liver play crucial roles involved in the process of whole-body energy homeostasis, and prolonged metabolic stress causes adipose tissue failure, inflammation, and adipokine release, resulting in an increase in lipid fluctuation in the liver and fatty liver [175]. Moreover, adipokines play a role in modulating insulin resistance, which is at the heart of obesity-related digestive diseases, such as gastroesophageal reflux disease (GERD), Barrett’s esophagus, esophageal cancer, colon polyps and cancer, NAFLD, viral hepatitis, cholelithiasis, gallbladder cancer, cholangiocarcinoma [176]. As mentioned, it has been suggested that a key contributor to the development of metabolic syndrome and its related pathophysiological effects is the chronic inflammatory condition that coexists with central obesity. However, it is crucial to emphasize the part adipokines play in this process specifically (Figure 2).

Considering the animal experiments that were conducted, through its metabolic and subsequent actions after weight loss, leptin reduces insulin resistance [177]. Additionally, it has been demonstrated that leptin administered intravenously to lean mice corrected hyperglycemia and hyperinsulinemia and enhanced insulin sensitivity [178]. Likewise, leptin is involved in the process of lipolysis, which releases triglyceride-based energy from white adipose tissue. White adipose tissue that is innervated by sympathetic nerves is stimulated by leptin. Chronic leptin injection stimulated the sympathetic tone of white adipose tissue, which led to a notable drop in epididymal fat weight without changes in food consumption, pointing to a role for leptin in the promotion of lipolysis [179]. In humans, extreme hyperphagia and obesity were caused by leptin-gene mutations and leptin-receptor abnormalities. In an observational trial with a 14-year follow-up period, higher leptin levels were linked to an increased incidence of metabolic syndrome [180]. Additionally, higher leptin levels were linked to a higher incidence of type two diabetes mellitus in a five-year prospective analysis of white men without diabetes [181]. Conversely, obese children with diabetes had low adiponectin levels, and a variation in the adiponectin gene resulting in a low adiponectin level was connected with the same disease [182], especially in elderly women [183]. Likewise, in obese women, a rise in adiponectin was linked to a reduction in hepatic insulin resistance [184]. Furthermore, higher fetuin-A levels have also been linked to insulin resistance in polycystic ovary syndrome and prepubescent children [185]. Regarding PAI-1 and lipocalin-2 levels, in comparison to the control group, the obese group had considerably higher outcomes, and it was positively correlated with markers of the metabolic syndrome such as higher (body mass index) BMI, skin-fold thickness, blood pressure, and LDL cholesterol level [186,187]. However, BMI, fat mass, and waist-circumference values were lower in patients with the highest visfatin levels among 350 obese women with metabolic syndrome [188].

Regarding resistin levels in experimental research, a probable mechanism for resistin’s proatherogenic activity has been presented. This suggested mechanism indicates that resistin reduces LDL-receptor (LDLR) expression in human hepatocytes in a PCSK9-dependent way. By altering PCSK9-induced LDLR expression, resistin can consequently influence serum lipid metabolism and cardiovascular disease. In relation to this, an observational study showed elevated serum resistin levels in obese people [189]. Likewise, plasma levels of omentin-1 and omentin-1 gene expression were significantly higher in lean subjects than in obese and overweight subjects [190]. In a prospective cohort study [116], the fasting asprosin levels of obese participants were considerably greater than those of normal-weight persons. Higher plasma asprosin levels were found to be related to poor glucose regulation and type two diabetes in a cross-sectional investigation [191]. Contrarily, those with type two diabetes had reduced adipocyte neuregulin four (NRG4) expression compared to those with adequate glucose management [192].

As has been shown, adipokine dysregulation affects people of all ages and genders. Our endocrine activities have a variety of effects on other organs through the release of adipokines that play essential roles in the regulation of energy balance and glucose homeostasis. This demonstrates a substantial association between the prevalence of metabolic disorders and the aforementioned variable. However, adipokines are also linked to cardiovascular disease and the disorder with the most severe repercussions associated with metabolic syndrome [193]. Given these functions, it has been suggested that adipokines may serve as diagnostic biomarkers or therapeutic targets for metabolic disease. At this time, at least 615 adipokines have been discovered as a component of the adipocyte secretome. In addition to classic adipokines, numerous research have been undertaken to find novel adipokines associated with metabolic syndrome as a result of the development of proteomics and metabolomics analysis methods [194].

## 10. Adipokines and Cancer

The overweight and obesity can increase the probability of developing certain types of noncommunicable diseases such as cardiovascular and metabolic diseases and different types of cancer [195]. In this sense, the types of cancer that are more likely to appear in people who are overweight are endometrial cancer, esophageal adenocarcinoma, gastric cardia cancer, liver cancer, kidney cancer, multiple myeloma, meningioma, pancreatic cancer, colorectal cancer, prostate cancer, gallbladder cancer, breast cancer, ovarian cancer, and thyroid cancer [196]. Thus, although the possible mechanisms by which excess weight may favor the development of these pathologies need more understanding, the scientific community has revealed that the main cause may be related to the proinflammatory adipokines produced in visceral fat and its close relationship with chronic inflammation processes [197]. In addition, obesity can cause a phenotypic change by inducing adipose tissue macrophages to stimulate increased production of proinflammatory cytokines such as tumor necrosis factor alpha, interleukin one beta, and monocyte chemoattractant protein, abrogating the activity of anti-inflammatory cytokines [198]. In relation to this, the hypertrophic expansion of adipose tissue increases local hypoxic processes, leading to the modification of different inflammation markers by the action of adipokines, which can increase carcinogenesis by altering cell differentiation and apoptosis [199]. In this sense, chronic inflammation in adipose tissue is stimulated and maintained by nuclear factor kB (NFkB) causing a pro-oncogenic environment [200]. Of the adipokines that have been characterized, leptin, resistin, visfatin, and apelin are abundant in the plasma of obese individuals, while adiponectin concentrations are minimal in obese individuals [201]. However, a recent meta-analysis established that adiponectin and leptin were the most studied, while interleukin-6, tumor necrosis factor-alpha, and resistin were relatively less studied [197].

According to leptin, it has been postulated as a potential overexpressed biomarker in breast tumors and metastatic lesions in women [202]. Leptin is a 16 kDa polypeptide produced mainly from white adipose tissue, and its plasma concentration directly correlates with the amount of that tissue [201]. Leptin binds to its receptors encoded by the LEPR gene, which is composed of six isoforms belonging to the cytokine receptor family [12]. The long isoform of this receptor is ObRb and it is expressed throughout the central nervous system (CNS) influencing appetite and metabolism. In this sense, when plasma leptin values are low in the fasting state, this can affect the actions carried out by the CNS [203]. After binding to this receptor, leptin promotes the recruitment and autophosphorylation of Janus kinase two (JAK2) and this causes signaling of the phosphoinositide 3-kinase (PI3K)/protein kinase B (Akt) and the mammalian target of the rapamycin (mTOR) pathways, which are responsible for cell survival and proliferation and promote cancer-cell growth [200]. Moreover, leptin influences the maintenance of the cancerous state through the signaling of signal transducer and activator of transcription three (STAT3) [204]. Thus, leptin may also increase the production of inflammatory proteins and growth factors that promote the growth and invasion of cancer cells [197]. Furthermore, leptin can also influence angiogenesis, stimulating the formation of new blood vessels, increasing their permeability, and facilitating the movement of cancer cells through them [12]. Regarding breast cancer, some studies suggest that leptin could be involved in the development and progression of this pathology. In this line, it has been shown that high levels of leptin can increase the proliferation of breast-cancer cells and decrease apoptosis in them [205]. In this sense, several meta-analyses have confirmed that breast-cancer patients had higher leptin levels [206]. Although the serum leptin level has been shown to positively correlate with breast-cancer risk in postmenopausal women, an inverse relationship has been reported in premenopausal women [207]. Regarding prostate cancer, studies have shown that men with moderate levels of leptin may have an increased risk of developing prostate cancer. In this line, Statin et al. [208] reported that intermediate levels, though not high, are the ones that are most related to the development of prostate cancer, establishing the existence of a critical fat mass related to a favorable internal environment for the development of this pathology. In addition, high plasma leptin levels may be correlated with the development of other types of cancer such as endometrial cancer, renal-cell carcinoma, thyroid cancers, and malignant melanoma [12].

According to adiponectin, it is a 30 kDa monomeric protein secreted by adipose cells that plays a variety of roles in the human body, including regulation of glucose and lipid metabolism, inflammation, and angiogenesis [209]. Adiponectin acts through its receptors AdipoR1, which is found mainly in skeletal muscle and hypothalamus, and AdipoR2, which is found mainly in white adipose tissue [12]. Even though this is the main element secreted by adipocytes, in obese people, adiponectin levels are inversely correlated with the amount of adipose tissue [197]. Thus, low levels of adiponectin in the body have been found to be associated with an increased risk of developing certain types of cancer, including colon cancer, breast cancer, prostate cancer, and endometrial cancer [209]. This is partly because adiponectin can act as a tumor suppressor, inhibiting the growth and proliferation of cancer cells, and reducing inflammation and angiogenesis that promote tumor growth [209]. Additionally, adiponectin has been found to improve insulin sensitivity and reduce insulin resistance, which may help prevent obesity and type two diabetes, known risk factors for many cancers [200]. Regarding breast cancer, the results of the different meta-analyses establish that high levels of adiponectin are associated with a lower risk of suffering from it when all women are included in the analysis. However, this association is more pronounced in postmenopausal women [210]. Moreover, decreased adiponectin levels have been associated with a higher histological grade of endometrial cancer. In this line, Endorgan et al. [211] reported after conducting a cohort study in Turkish women that patients with endometrial cancer had significantly lower serum adiponectin levels than the controls, indicating the risk factor for the development of endometrial cancer in postmenopausal women with a decrease in the values of this adipokine. Consistent with the progression of colorectal cancer, adiponectin, and Pai-1 have been postulated to be mediators of this pathology [212]. In this sense, Gonullu et al. [212] (reported that in addition to being associated with insulin resistance, low adiponectin levels indicate a poor prognosis along with carcinogenesis in patients with colon cancer.

Regarding resistin, it is a 12 kDa protein member of the family of cysteine-rich proteins which is secreted by mononuclear cells and adipocytes [213]. Although the receptor for this adiponectin has not been definitively identified, it has been speculated that Toll-like receptor four (TLR4) and adenylyl cyclase-associated protein one (CAP1) may play a role. [214]. According to resistin levels, the highest contributors are monocytes and macrophages associated with adipose tissue [215] and elevated levels of this adipokine in plasma are associated with the development of certain pathological states such as visceral obesity, coronary artery disease, lung disease, various malignancies, and critical illness [216]. In relation to cancer development, high levels of serum resistance have been reported in many types of cancer and not only those that are dependent on obesity [201]. Moreover, Lerkhe et al. [217] hypothesized that resistin could be one of the adipokines that may have the greatest weight in the development of different types of cancer since the levels of resistin were more likely related to the inflammatory status of the individual. In this sense, several meta-analyses [206,218] have reported that high levels of resistance were highly related to the incidence of breast, endometrial, and colorectal cancer. In this line, it has been proposed that resistin can activate different signaling pathways such as mitogen-activated protein kinase (MAPK), Akt, and P1K3, increasing the proliferation of cancer cells and activating the NFkB signaling pathway, which can increase levels of IL-6-inducing metastatic processes [219]. Regarding breast cancer, Gui et al. [206] reported in their meta-analysis that higher resistance levels were associated with breast cancer, although there was no significant association between resistance levels and menopausal status. In this line, it has been established that in breast cancer, resistin induced the phosphorylation of proto-oncogene tyrosine-protein kinase Src (c-Src), protein phosphatase 2A (PP2A), and protein kinase C alpha (PKCα), among others, promoting cancer progression [220]. In relation to multiple myeloma, Dalamaga et al. [218] reported that lower serum adiponectin and resistin levels were associated with increased risk in patients diagnosed with this type of cancer compared with the controls. Furthermore, Danese et al. [221] analyzed in their study 40 patients with colon cancer who reported higher levels of resistance with respect to the controls, indicating the association between this adipokine and the development of a greater inflammatory processes during the disease.

Regarding visfatin, it is also known as Nampt or pre-B cell colony-enhancing factor (PBEF) and is a 52-kDa protein, that is produced by the NAMPT gene [200]. High levels of visfatin in plasma have been shown to positively correlate with obesity and, above all, with the amount of visceral fat [222]. Thus, recent studies indicate that visfatin may be more expressed in macrophages infiltrating adipose tissue, suggesting that macrophages release visfatin in response to inflammatory signals rather than adipocytes [223]. Like other adipokines, visfatin promotes proinflammatory, proliferative, antiapoptotic, and proangiogenic effects [224]. In this line, the main mechanism of action of visfatin for the development of carcinogenesis is the production of inflammatory cytokines, such as Il-6, TNF, or hypoxia-inducible factor 1α (HIF-1α). However, it has also been reported that a second mechanism could be the expression of stromal cell-derived factor one (SDF-1) that promotes the survival and migration of cancer cells [225]. In this line, Mohammadi et al. [222] reported in their meta-analysis that high levels of this adipokine in plasma indicated a strong association with the risk of various types of cancer, as well as being a good biomarker for detecting cancer early. Specifically, several studies have reported the relationship between high levels of visfatin and colorectal cancer [226], breast cancer, or postmenopausal breast cancer [227]. Furthermore, elevated levels of visfatin in breast cancer are associated with more malignant tumor behavior as well as poor patient survival [228].

Apelin is a 9-kDa peptide and is the ligand of the G protein-coupled APJ receptor [201]. It has been reported that this adipokine is involved in different metabolic processes such as insulin regulation and sensitivity [229]. Similar to the rest of the adipokines mentioned above, high levels of apelin have been observed in obese people, although other studies have reported high levels of this adipokine in other pathologies and healthy subjects [201]. In relation to cancer, it has been observed in animal models that when this adipokine is overexpressed in cancer cells, it causes an increase in vascularization and an increase in tumor size [230]. In addition, it has been described that apelin induces carcinoma proliferation and cell migration under hypoxic conditions under HIF-1α signaling [231]. However, high plasma levels of apelin have also been correlated in human models in lung, colon, gastroesophageal, hepatocellular, and breast cancer, among others [232], increasing the risk of developing this type of pathology. In this line, Altinkaya et al. [233] reported significantly higher levels of apelin in obese women with endometrial cancer than in normal-weight women with the same disease, indicating that elevated circulating levels of this adipokine are a risk factor for increasing problems associated with this disease. According to breast cancer, several immunohistochemical studies have reported higher levels of apelin in this type of pathology [234]. Some have even shown this increase in postmenopausal women with a reduction in these values after treatment with aromatase inhibitors [235]. In this line, it has been postulated that apelin acts by signaling the extracellular signal-regulated kinase 1/2 (ERK1/2) and PI3K–Akt pathways promoting lymphangiogenesis, tumor neoangiogenesis, promotion of cell proliferation, and invasion [236].

## 11. The Role of Microbiota in Adipokines

The human body is home to trillions of microorganisms that collectively make up the gut microbiota. Recent research is suggesting that the gut microbiota could be one of the neglected environmental factors implicated in the regulation of the myokine-adipokine profile. These microorganisms play a crucial role in maintaining the health of the digestive system, immune system, and even the brain [237]. Recent research has shown that the gut microbiota may also play a role in the regulation of adipokines, which are signaling molecules produced by adipose tissue, that are involved in regulating metabolism and inflammation in the body [13]. Its metabolites signaling in the G protein-coupled receptors GPR41 and GPR43 present in adipose tissue stands out, as well as the influence on intestinal incretins that modulate energy metabolism and central-intake regulation [238]. Along these same lines, the microbiota regulates innate and adaptive immunity, and influences local and systemic mucosal responses; therefore, it influences chronic inflammation associated with obesity and insulin resistance. Thus, adipokines are important regulators of energy metabolism, with some playing a role in increasing energy expenditure and others involved in storing energy as fat. Among the most well-known adipokines are adiponectin, which has insulin-sensitizing and anti-inflammatory effects, and leptin, which regulates appetite and energy balance. Dysregulation of adipokine production and release has been linked to the development of obesity, type two diabetes, and other chronic diseases. In this line, studies have shown that the composition and diversity of the gut microbiota may influence the production and release of adipokines [239].

The mechanisms by which the gut microbiota influences adipokine production and release are not yet fully understood. However, it is thought that the gut microbiota may influence the expression of genes involved in adipokine synthesis and secretion, as well as contribute to the development of chronic low-grade inflammation that is often associated with obesity and metabolic dysfunction [240]. Additionally, certain bacterial metabolites produced by the gut microbiota, such as short-chain fatty acids, may also play a role in regulating adipokine production and release.

Furthermore, a growing body of research indicates that changes in the gut microbiota’s composition and/or function under pathological circumstances, sometimes referred to as “dysbiosis,” may be crucial in the development of a number of metabolic disorders, including obesity, type two diabetes, liver disease, cancer, and even neurological disorders [13]. Changes in the composition of the gut microbiota have been associated with gut-barrier dysfunction (e.g., decreased mucus-layer thickness, disruption of the tight-junction proteins, and decreased secretion of antimicrobial peptides), which allows pathogen-associated molecular patterns (PAMPs) to cause abnormal host immune response and low-grade inflammation. In bile acid profiles, dysbiosis has also been associated with reduced secretion of intestinal peptides, lower synthesis of short-chain fatty acids (SCFAs), and greater amounts of branched-chain amino acids. Given the significant role that the gut microbiota plays in sustaining a decent level of health and wellbeing, its manipulation is seen as a key strategy for either preventing or treating metabolic illnesses linked to dysbiosis [15]. The data is unequivocal in stating that the gut bacterial profiles of obese and lean animals differ in composition and functioning [241]. For instance, while analyzing the MI of male Sprague–Dawley (SD) rats fed high-fat milk for 19 days to produce obesity, a rise in Lactobacillus (LAC) and a decrease in Bacteroides (BAC) proportions were recorded compared to healthy controls [242]. For instance, while analyzing the microbiota of male Sprague–Dawley (SD) rats fed high-fat milk for 19 days to produce obesity, a rise in Lactobacillus (LAC) and a decrease in Bacteroides (BAC) proportions were recorded compared to healthy controls [242]. It is interesting to note that levels of LAC were positively connected with the animals’ increasing adiposity and energy intake, proving that both factors are impacted by excessive energy consumption. In a similar vein, De la Serre et al. reported a drop in overall bacterial density and a relative proportion of Bacteroidales/Clostridiales in rats with an obesity-prone phenotype, confirming the shift. It is interesting to note that the latter showed a rise in Toll-like receptor four (TLR4) activation and a decline in intestinal alkaline phosphatase, which are factors associated with ileum inflammation and lipopolysaccharide (LPS) regulation, respectively [243].

Hamilton et al. then presented crucial data to corroborate the hypothesized causal relationship between microbiota alterations and the start of obesity pathogenesis [244]. The authors identified probable early events that might lead to obesity and its associated comorbidities, such as the dysregulation of IL-10 and transcellular flow at the level of the large intestine. In this regard, intestinal permeability dysfunction and inflammation have been hypothesized as microbiota-mediated events linked to the beginning of obesity pathogenesis in animal models [245]. Adipose tissue and the central nervous system (CNS) have both been shown to be affected by gut bacteria systemically, which alters body composition by controlling intake and energy metabolism [246].

Each of the modifications is linked to or generated from an obese microbiota, according to all the aforementioned pathways, and its continuous dysregulation is what causes both the maintenance of the obesity state and its repercussions [247]. Therefore, there is a bidirectional relationship, a vicious feedback loop in which the cause of the dysregulation is focused on the adipocyte, and for this, we have to go in the short term to a caloric surplus and poor nutrition, and in the long term, to the appearance of overweight and obesity.

Regarding the modulation of inflammation at the level of the adipose tissue. Virtue et al. demonstrated an association between the concentration of bacterial metabolites derived from intestinal tryptophan and the expression of miR-181, which promoted inflammation of the target AT and, consequently, obesity [248]. On the other hand, the signaling of SCFAs has been reported in GPR43 receptors expressed by immune cells, including macrophages. The relationship with adipose tissue would be provided by the recruitment of macrophages from the bloodstream that is affected precisely by SCFAs and IDO [249]. Studies have defined that precisely these macrophages can capture LPS through their TLR4 membrane receptors, promoting the conversion of the M2 phenotype to the M1 phenotype, and with it, the consequent secretion of IL-b1 and TNF-a, both proinflammatory cytokines. In turn, a decrease in TLR4 receptors decreases inflammation by promoting the M2 phenotype in macrophages [250]. These data suggest that macrophages would be responsible for driving chronic low-grade inflammation in adipose tissue, precisely by promoting a sustained increase in proinflammatory cytokines and a decrease in anti-inflammatory molecules. Considering all of the above, intestinal dysbiosis could be involved not only in the event that triggers inflammation of adipose tissue but also in maintaining the proinflammatory environment at the adipose and, above all, the systemic level, preventing anti-inflammatory processes that could mitigate its action.

Yet, the adipose tissue itself and its relation to microbiota was further discussed by authors such as Suarez-Zambrano et al., who reported that intestinal bacteria have the ability to modulate adipogenesis at the level of WAT [251]. The authors suggest that the depletion of microbiota from two different methods promoted the browning of adipose tissue at the subcutaneous inguinal and visceral perigonadal levels, via infiltration of eosinophils and by the action of M2 macrophages. In the same line, Jian et al. reported that the depletion of microbiota affected the thermogenesis of brown adipose tissue (BAT), a critical mechanism of this tissue to increase the energy expenditure of the organism, and that precisely confers protective properties for the obese-gene phenotype [252]. Interestingly, the alteration would be attenuated by butyrate, thus establishing a modulating role for the microbiota in the critical functions of the adipose tissue. Regarding visceral adipose tissue (VMF) and the link with microbiota, a study in the Twins UK cohort managed to determine the association of seven bacterial genera with FMD content [253], with the genus Blautia being the most prominent for being identified as a hereditary component.

As will be discussed later in the following points, there are interventions at the nutritional or physical exercise level that can modulate the intestinal microbiota, and this, in turn, is related to the lipid adipocyte profile, and the adipokines that are generated. In this line, regarding WAT, interesting strategies have been published for its modulation of microbiota. Among them, intermittent fasting every two days in animal models proved to be beneficial for inguinal WAT browning, inducing metabolic improvements, as well as increasing energy expenditure from lipid oxidation, with only 15 fasting cycles [254]. The browning achieved was developed independently of BAT activation, from microbiota-mediated acetate and lactate. As confirmed by Jockens et al., describing the role of SCFAs in intracellular lipolysis in hMADS adipocytes [255]. On the other hand, the regulation of WAT and its influence on microbiota-mediated insulin resistance could be explained by the effect of tryptophan-derived metabolites on the expression of miR-181 in WAT [248].

Thus, the gut microbiota is one of the neglected factors implicated in the regulation of the myokine-adipokine profile. The understanding of the pathophysiology of obesity, the physiology of adipose tissue, and the interactions with the bacterial profile of the host, as well as its effect on it, are essential in health (Figure 3).

## 12. The Role of Nutrition in Adipokines

Nutrition plays a significant role in the pathogenesis of adult diseases, especially metabolic diseases. The influence of nutritional factors on the development of obesity and its associated metabolic complications in later adulthood has sparked considerable interest in the field of infant nutrition [256].

As has been established previously, adipose tissue is a major endocrine organ that plays a crucial role in the development of metabolic diseases via adipokines [257]. Adipose tissue enlargement causes an imbalance in adipokine secretion in obese individuals. In these diseases, circulating levels of several proinflammatory adipokines, such as leptin and TNF- α, are elevated, followed by a decrease in adiponectin expression, which contributes to the characteristic low-grade chronic inflammatory status. AT inflammation is a potential link between obesity, insulin resistance, and type two diabetes [258]. Additionally, increased fat mass is associated with a state of chronic inflammation, macrophage infiltration of WAT, and abnormal production of adipokines and cytokines in obese individuals. It is presently debatable how specific nutrients affect the production and secretion of adipokines and WAT-derived cytokines [259].

Regarding leptin levels, they correlate with body-mass index and body-fat percentage in animal and human models. Obese patients exhibit “resistance” to leptin, which is characterized by a decreased tissue response to leptin [260,261]. Obesity is exacerbated by leptin resistance, which then contributes to an increase in leptin levels, creating a vicious circle known as “leptin-induced leptine resistance” [260]. Various diets and dietary supplements can affect leptin expression. In obesity treatments, normalizing leptin sensitivity has been proposed as a strategy to increase weight loss and prevent weight regain [261]. Reducing the amount of adipose tissue and leptinemia simultaneously has been demonstrated to restore leptine sensitivity in animal models [260]. Although leptin is a central component of the metabolic changes that accompany weight loss and can trigger body-weight restoration [262], it may be incorrect to assume that reducing leptine levels alone will improve weight loss.

Thus, the adiposity- and leptin-reducing effects of dietary interventions such as energy restriction and fasting may help restore sensitivity to this hormone by modulating the biological mechanisms underlying leptine resistance [263]. In this regard, numerous studies have evaluated the short- and long-term effects of energy-restricted diets on leptin levels in humans. In a case-control study, Boden et al. found that the absence of leptin changes during fasting, when baseline insulin and glucose levels remained unchanged, suggested that insuline and/or glucosa may play a role in regulating leptine release [264]. Although leptin concentrations in fasting subjects decreased significantly during energy restriction and subjects exhibited compensatory behavior during subsequent food intake in pleasant-eating subjects, no correlation was found between decreasing fasting, leptine concentrations, and calorie compensation, as determined by Mars et al. [265]. Wadden et al. discovered that the degree of calorie restriction, but not weight loss, significantly contributed to the variation in leptin change; on the other hand, long-term changes in leptin, when subjects had increased their calorie intake, were more strongly related to weight changes; serum leptin and body fat remained highly correlated after weight loss [266]. In a meta-analysis, Varkaned et al. found a significant effect of fasting and energy-restricted diets on leptin concentrations, though no significant effect of fasting and energy restriction on adiponectine concentrations. However, restricted energy regimes significantly increased adiponektin concentrations [267].

Moreover, a number of studies have demonstrated that supplementation with probiotic and prebiotic foods improves glucose parameters and leptin concentrations in patients with obesity [268], diabetes [269], and nonalcoholic fatty liver disease [270]. Lactobacillus rhamnosus, Lactobacillus acidophilus, and Bifidobacterium bifidumi restored intestinal microbiota and enhanced leptin resistance in obese mice [271]. Future research should clarify the role of prebiotics and probiotics in leptin synthesis in humans, as prebiotics and probiotics are likely to be more effective when combined with lifestyle modifications [272].

According to the current data, omega-3 polyunsaturated fatty acids increase leptin levels in obese patients, whereas they decrease leptine levels in lean subjects [273]. The combination of lifestyle changes and supplementation with n-3 fatty acids, eicosapentaenoic acid (EPA), and docosahexaenoic acid (DHA) reduces inflammatory and metabolic damage in obese women but does not affect leptin levels [274]. In contrast, lifestyle interventions with increased omega-3s reduce insulin resistance, leptin concentrations, and endothelial dysfunction in obese adolescents [275].

Dried fruits contain polyphenols, and it is known that leptin-induced damage is associated with oxidative and inflammatory stress. In obese animals fed a diet rich in carbohydrates and fats, nut oil increases insulin sensitivity and antioxidant capacity while decreasing leptin concentration [276], and supplementation with mixed nuts promotes satiety and reduces leptine concentration in obese and overweight adults [277].

In addition to leptin, adiponectin’s potential effects on caloric restriction have been the subject of numerous research articles. The primary function of APN, also known as “ inanition protein,” is energy homeostasis [272]. APN improves glucose intake and prevents gluconeogenesis and fatty-acid accumulation by activating AMPK signaling, whereas leptin has the opposite effect. Low APN concentrations and an altered ratio between leptin and APN are associated with BMI, altered insulin signaling, and an inflammatory state in obese patients [278]. Reduced levels of APN are strongly associated with cardiovascular disease in animal models, with increased expression of anti-inflammatory genes, including eNOS and IL-10, and suppression of proinflammatory gene expression (TNF α-and IL-6) [279].

In animal obesity models, calorie restriction reduces left ventricular hypertrophy and diastolic dysfunction while increasing adiponectin [280]. Calorie restriction improves insulin resistance by increasing adiponectin production by adipocytes [281]. In addition, this nutritional approach increases the concentration of adiponectin in obese patients, which promotes weight loss [282]. In addition, the increased benefits of the combined therapy of caloric restriction and exercise amplified the effect of caloric restriction on adiponectin expression [283].

The identification of new therapeutic agents or treatments that have a positive effect on the expression of these molecules could have a significant positive impact on the management of obesity and its cardiovascular complications [18].

## 13. The Role of Physical Activity in Adipokines

Regarding physical exercise, numerous studies have established that people who are more physically active or have better aerobic fitness, tend to have more favorable adipokine profiles and lower levels of oxidative stress [236]. In this line, it has been reported that increased physical activity improves adipokine levels and reduces all-cause mortality [284]. In addition, a direct relationship has been reported between the practice of different sports modalities with increased levels of anti-inflammatory cytokines and decreased inflammatory cytokines [285]. However, studies that have observed improvements in physical exercise on adipokine levels have also observed concomitant improvements in body composition associated with it, which could confound data regarding the direct effects of exercise on these variables [286]. In this sense, some studies have questioned the improvement of adipokine levels due to physical exercise independently of weight loss [287].

The potential of physical exercise to reduce the levels of proinflammatory cytokines or adipokines has been well documented. However, this improvement is directly related to the type of exercise that is performed, its intensity, and its duration [288]. In this sense, it has been shown that performing combined activities (for example, aerobic exercise and resistance training) report better results than both modalities separately in controlling and reducing the levels of these parameters [289].

Leptin has been shown to directly influence fatty-acid metabolism and the endocrine axis, in addition to influencing skeletal muscle fatty-acid metabolism [290]. In this sense, the effect of physical exercise on leptin concentrations is currently controversial, since different investigations have reported that physical exercise can result in reductions depending on the duration and calorie expenditure, while others did not report changes in leptin concentrations.

Thus, Bouassida et al. [291] did not report a decrease in plasmatic leptin levels in a sample of 5 physically active men and 12 women after performing 45 s of exercise at an intensity of 120% of maximum aerobic power peak. In this same line, Zoladz et al. [292] also did not observe changes in leptin concentrations in a sample of eight physically active men after performing a maximal and incremental cycle ergometer protocol without fasting and a submaximal incremental cycle ergometer protocol under fasting conditions. Thus, Kraemer et al. [293] indicated that exercise protocols that last less than 60 min or that generate a caloric expenditure of less than 800 kcal are not optimal for modifying leptin concentrations. Furthermore, some investigators have reported that postexercise cortisol release may affect leptin concentrations. However, Zaccaria et al. [294] compared the effects of three different competitive endurance modalities in 45 trained men. In this sense, the reduction in leptin levels was only observed in the alpine ski and ultramarathon modalities, which had an approximate energy expenditure of 5000 and 7000 kcal, respectively. In this sense, it was hypothesized that activities of very long duration or those that generate a large energy expenditure can modify the daytime leptin secretion rhythm, causing these decreases and revealing an important relationship between leptinemia and energy expenditure [290].

Regarding strength training, Zafeiridis et al. [295] reported a decrease in serum leptin levels 30 min after performing three acute resistance training sessions with different protocols (hypertrophy, strength, and muscular endurance) compared to their baseline. However, they found no significant differences between protocols. In this line, Racil et al. [296] reported that high-intensity interval training (HIIT), together with plyometric training, caused an improvement in lean body mass and a decrease in leptin concentrations and the leptin/adiponectin ratio than plyometric training alone in young obese females. According to this, the improvement of leptin levels is associated with an improvement in the state of health of people with obesity since leptin increases the oxidation of glucose, normalizing the rate of glycogen synthesis [297].

In relation to adiponectin, a moderate association between this marker and the traditional markers of metabolic health has been reported in [298], showing an inverse relationship with insulin resistance [299]. However, unlike leptin, adiponectin values are lower in obese people [300]. The different studies carried out in children or adolescents with obesity indicate that the adiponectin response to exercise depends on the different hormonal changes associated with sex in the pre- or postpubertal age [19]. In this sense, Fazelifar et al. [301] observed a decrease in adiponectin levels compared to baseline in both study groups after 12 weeks of concurrent training or the maintenance of daily usual activities (control group). However, after four weeks of detraining, adiponectin levels did not increase in the experimental group despite gradual deconditioning. Thus, Kim et al. [302] showed an improvement in triglyceride and insulin sensitivity and increased adiponectin levels in obese Korean male adolescents. However, these improvements were not correlated with an improvement in different inflammatory markers such as high-sensitivity C-reactive protein (hs-CRP), interleukin (IL)-6, and tumor necrosis factor (TNF)-α. However, Vasconcellos et al. [303] reported no improvements in the level of adiponectin in obese Brazilian adolescents after 12 weeks of a recreational soccer training program even though the experimental group increased the level of different markers of cardiovascular health and physical fitness.

In adults, it has been reported that improvements in adiponectin levels after a physical exercise program are more related to the level of the physical condition [304] than to the modality of exercise that is carried out [305]. In this line, Jurimae et al. [306] reported a significant decrease in adiponectin levels after an acute training session in highly trained rowers. However, 30 min after finishing the training, the adiponectin values increased by 20% with respect to the basal levels. However, Hulver et al. [307] reported no changes in adiponectin levels after a six-month exercise protocol performed four days/wk for 45 min at 65–80% peak O2 consumption in sedentary healthy subjects. In contrast, Ring-Dimitriou et al. [308] reported an increase in adiponectin levels together with an improvement in cardiorespiratory fitness in men with a predisposition to suffering from metabolic syndrome.

Resistin is the most studied adipokine in obese individuals and it has been reported that high levels of resistin in plasma are correlated with poor physical condition [309]. This parameter causes insulin resistance in liver tissue and plays a role in inflammatory and autoimmune processes [310]. In children, Kelly et al. [236] reported that eight weeks of aerobic training did not produce a reduction in weight or an improvement in resistin levels, indicating that physical exercise in isolation, with no weight loss, is not effective in modifying this adipokine. In contrast, Ounis et al. [311], showed a correlation with enhancement in lipid oxidation and resistance among other proinflammatory adipokines in obese female adolescents. However, a meta-analysis established that exercise did not reduce resistance concentrations in pediatric obesity due to the small sample size included in both studies [312]. In this line, Jamurtas et al. [313] conducted an acute aerobic training session at 65% of VO_2_max for 45 min, finding no reduction in resistin levels at four points in time (pre, post, post-24 h, and post-48 h) in overweight men. In diabetes type two patients, high-intensity combined exercise training (aerobic and resistance), in addition to daytime physical activity, is required for achieving a significant decrease in resistin serum levels [314]. In this vein, another study showed an important reduction in resistin levels after 12 weeks of resistance training (10.8%) and combined training (7.8%) in comparison with aerobic training and stretching control conditions in which this reduction was less [315]. However, data on resistin levels in healthy individuals in relation to exercise are scarce [316].

Visfatin is an adipokine expressed in visceral adipose tissue and has been shown to exert an insulin-mimetic effect [317]. Furthermore, it can act as an autocrine, paracrine, and endocrine mediator, participating in the regulation of a wide variety of physiological functions, including cell proliferation or contributing to glucose-related conditions and obesity [312]. In obese female adolescents, Lee et al. [318] reported a reduction in plasma visfatin levels after 12 weeks of aerobic training with an energy expenditure between 1200 and 1600 kcal per week. In this line, 12 weeks of combined training (aerobic and resistance training) performed 65 min a day for five days a week reduced plasma visfatin values (13.6 ± 12.0 to 7.7 ± 7.9 ng/mL) in nondiabetic obese Korean women [319]. In accordance, Seo et al. [320] reported a significant reduction of visfatin levels accompanied by an improvement in body composition and markers of metabolic syndrome in obese middle-aged women after 12 weeks of combined training (aerobic and resistance training) three days a week with 60 min a day. In contrast, increases in plasma visfatin levels, together with increased plasma glucose and insulin concentrations have been observed after high-intensity exercise (repeated sprint-ability training) in physically healthy men, indicating an increase in the ability of the tissues for glucose uptake and glycogen restoration after exercise [321]. However, these parameters returned to their baseline values 45 min after exercise. In type one diabetes, Haider et al. [32] reported that four months of supervised aerobic training performed two to three times per week was sufficient to reduce plasma visfatin levels in patients with type one diabetes, maintaining this reduction eight months after cessation of the training protocol. In type two diabetes, combined exercise (aerobic and resistance training) caused a significant reduction in visfatin levels compared to other training protocols (aerobic or resistance training alone) after 24 weeks of experimental design [322].

## 14. Practical Applications

A better understanding of these important cytokines and their effects on the organisms inside the body would be possible with the knowledge gained from this review. Although future studies should concentrate on the examination of adipokine secretion, as this could aid in our understanding of the full spectrum of obesity-related diseases. This knowledge sheds light on nonpharmacological treatments such as exercise or nutrition as promising innovative treatment modalities. In order to differentiate between cause and effect and compensatory or operational function, it may be possible to implement specific exercise protocols. Dietary interventions may also have a significant positive impact on adipokine concentrations, which is largely explained by a decrease in fat mass. Early detection could aid in our understanding of the role and molecular targets of the most recent adipokines.

## 15. Conclusions

There is a growing body of evidence that suggests that adipokines play crucial roles in the regulation of inflammation state. In particular, the information provided in this review indicates that in a state of inflammation, dysfunctional adipose tissue leads to the dysregulation of adipokines, which in turn causes multiple diseases. In Table 1 and Table 2, the most pertinent references with conclusive results are presented.

Leptin and adiponectin play important roles in this regulation, with increased leptin and decreased adiponectin being the primary causes of cancer, cardiovascular and metabolic diseases, as well as problems in the gut microbiota and, consequently, mental health as well as immunity.

Although the levels of adipokines during exercise are somewhat debatable, adiponectin levels have been shown to increase with exercise, leading to the prediction of decreased insulin resistance. On the other hand, visfatin and leptin have been associated with obesity; therefore, increased physical activity is also proposed as a factor in reducing obesity. Consequently, people who are more physically active or practice aerobic training and highly intense exercise tend to have more favorable adipokine profiles and lower oxidative stress levels. Combining these activities with resistance training decreases levels of proinflammatory cytokines or adipokines and improves, consequently, better insulin sensitivity and glucose metabolism.

Regarding nutrition, a fat-based diet affects the gut microbiota, which in turn can influence the production of adipokines. Although evidence has shown that leptin parameters do not change with calorie restriction, other aspects such as probiotic and prebiotic intake do promote leptin reduction. Additionally, a diet rich in fruits, vegetables, and fiber has been linked to higher levels of anti-inflammatory adipokines. Consequently, exercise and a healthy diet are hypothesized to be the factors that can keep proinflammatory adipokines in control.

## Figures and Tables

**Figure 1 biomedicines-11-01290-f001:**
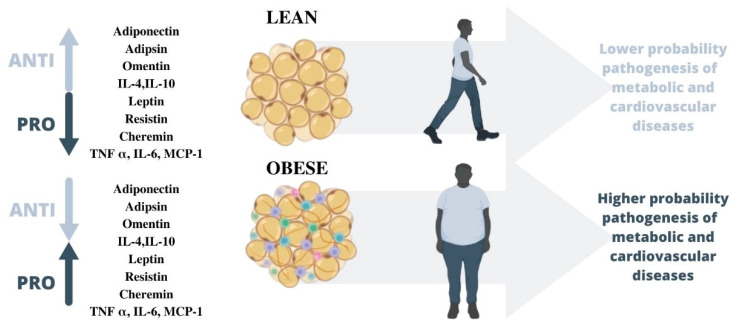
The behavior of adipokines in relation to the adipose tissue status of an individual. In healthy adipose tissue, anti-inflammatory adipokines increase while proinflammatory adipokines decrease, whereas the opposite is true in obese adipose tissue, increasing the likelihood of cardiovascular and metabolic diseases.

**Figure 2 biomedicines-11-01290-f002:**
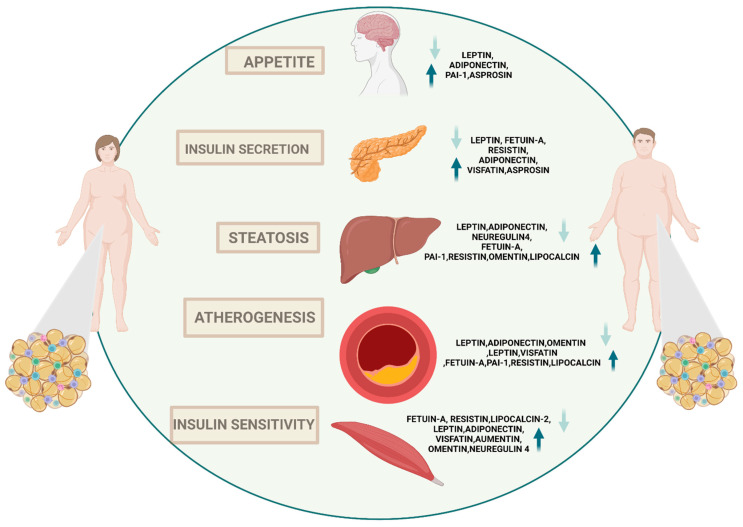
Metabolic illness is described by the altered behavior of adipokines. A rise in proinflammatory adipokines and a fall in anti-inflammatory adipokines can have an effect on appetite and other factors. Liver steatosis and cardiovascular atherogenesis are both possible. In addition, insulin production from the pancreas is disrupted and insulin sensitivity is compromised.

**Figure 3 biomedicines-11-01290-f003:**
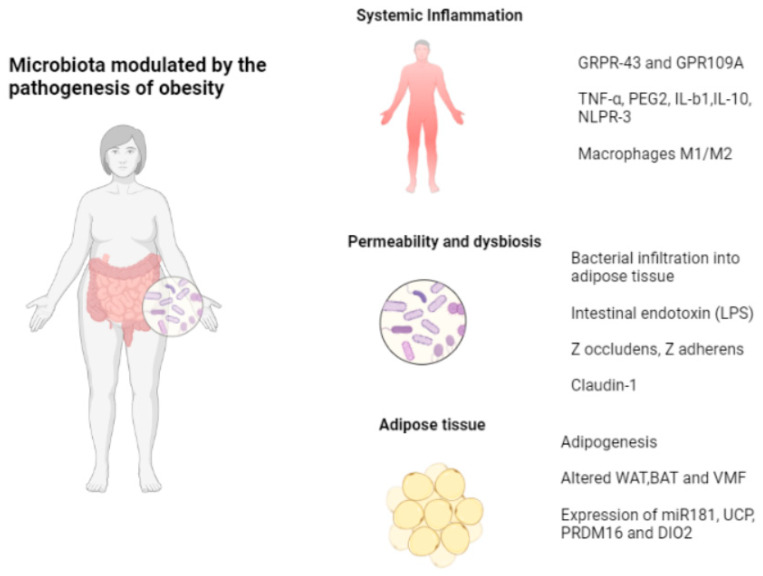
The pathogenesis of obesity induces a low-grade inflammation that generates a dysregulation in the adipose tissue and microbiota, which affects our immune system and results in an increase in intestinal permeability.

**Table 1 biomedicines-11-01290-t001:** Summary of the most relevant conclusions of each section (from 4 to 8).

Authors and Year	Study Title	Aim of Study	Main Outcomes	Section
Tilg et al. (2006) [46]	Adipocytokines: Mediators Linking Adipose Tissue, Inflammation, and Immunity	Understand adipocyte-derived mediators.	Adipocytokines—mediators are mainly synthesized by adipocytes in white adipose tissue. Adiponectin and leptin are the two most abundant adipocytokines.	Adipokines and inflammation
Ohashi et al. (2014) [66]	Role of Anti-Inflammatory Adipokines in Obesity-Related Diseases. Leptin in inflammation and autoimmunity	Understand the role of the anti-inflammatory adipokines.	Adiponectin exerts beneficial actions on obesity complications.	
La Cava (2018) [59]		Analyzes the known implications of leptin in multiple inflammatory conditions, including autoimmune diseases.	Leptin contributes to the generation and maintenance of low-grade inflammation.	
Mitsuhashi et al. (2007) [80]	Adiponectin Level and Left Ventricular Hypertrophy in Japanese Men	Analyze the relationship that might exist between the serum adiponectin level and electrocardiographically diagnosed left ventricular hypertrophy (ECG-LVH)	Adiponectin concentration was inversely and independently associated with ECG-LVH.	Adipokines and cardiovascular disease
Khafaji et al. (2012) [91]	Elevated Serum Leptin Levels in Patients with Acute Myocardial Infarction; Correlation with Coronary Angiographic and Echocardiographic Findings	Assess the relationship between serial serum leptin levels in patients with acute myocardial infarction (AMI)	There was a trend for an increase in the mean serum leptin levels with an increasing number of diseased vessels.	
Hassan-Ali et al. (2011) [121]	Serum Adiponectin and Leptin as Predictors of the Presence and Degree of Coronary Atherosclerosis	Determine the relationship between serum adiponectin and leptin with the presence and degree of coronary atherosclerosis.	Both serum adiponectin and leptin might play an important pathogenic role not only in the occurrence of coronary artery disease patients.	Adipokines and atherosclerosis
Satayesh et al. (2021) [135]	The Possible Mediatory Role of Adipokines in the Association between Low Carbohydrate Diet and Depressive Symptoms among Overweight and Obese Women	Explore the possible mediatory role of adipokines Galectin-3, transforming growth factor-beta (TGF-β), and endothelial plasminogen activator inhibitor (PAI-1) in the association between low carbohydrate diet (LCD) and depressive symptoms.	Higher adherence to Low Carbohydrate Diet is probably associated with a lower prevalence of depressive symptoms in obese adults through the mediatory role of adipokines.	Adipokines and mental disease
Yu et al. (2021) [323]	Gut Hormones, Adipokines, and pro- and Anti-Inflammatory Cytokines/Markers in Loss of Control Eating: A Scoping Review	Synthesize research that has investigated these biomarkers with loss of control eating.	The use of glucagon-like peptide 1 analog was able to decrease binge eating.	Adipokines and eating behaviors

**Table 2 biomedicines-11-01290-t002:** Summary of the most relevant conclusions of each section (from 9 to 13).

Authors and Year	Study Title	Aim of Study	Main Outcomes	Section
Ma et al. (2020) [183]	Lower Levels of Circulating Adiponectin in Elderly Patients with Metabolic Inflammatory Syndrome: A Cross-Sectional Study.	Assess whether circulating adiponectin can be used as an indicator for the metabolic inflammatory syndrome (MIS) in elderly adults.	Lower adiponectin levels in serum are associated with MIS	Adipokines and metabolic diseases
Sánchez-Jiménez et al. (2019) [202]	Obesity and Breast Cancer: Role of Leptin	Know a better knowledge of the molecular mechanisms that mediate leptin action may be helpful to understand the underlying processes which link obesity to breast cancer in postmenopausal women.	The activation of leptin signaling results in the concurrent activation of multiple oncogenic pathways leading to increased proliferation.	Adipokines and cancer
Stattin et al. (2001) [208]	Leptin Is Associated with Increased Prostate Cancer Risk: A Nested Case-Referent Study	Study the association between obesity and prostate cancer	A critical fat mass related to an interior milieu favorable for prostate cancer development seems to exist because intermediate but not high leptin levels are related to prostate cancer risk.	
Hong et al. (2016) [252]	Butyrate Alleviates High Fat Diet-Induced Obesity through Activation of Adiponectin-Mediated Pathway and Stimulation of Mitochondrial Function in the Skeletal Muscle of Mice	Explore whether the adiponectin-mediated pathway is involved in the anti-obesity action of butyrate	Short-term oral administration of SB can alleviate diet-induced obesity and insulin resistance in mice through the activation of the adiponectin-mediated pathway	The role of microbiota in adipokines
Pigatto et al. (2021) [324]	Gastrointestinal involvement in systemic sclerosis: Pathogenic role of the gut microbiome, cytokines, and adipokines	Evaluate serum levels of adipokines and cytokines involved in the pathogenesis of Systemic Sclerosis and their role.	Serum levels of IL12 and IL10 were found to correlate with specific bacteria alterations.	
Mendoza-Herrera et al. (2021) [261]	The Leptin System and Diet: A Mini Review of the Current Evidence	Review the literature on the relationship between diet and leptin, which suggests that addressing leptin resistance through dietary interventions can contribute to counteracting obesity	The link between nutritional components and leptin resistance, as well as research indicating that this condition is reversible, emphasizes the potential of diet to recover sensitivity to this hormone.	The role of nutrition in adipokines
Jadhav et al. (2021) [288]	Effect of Physical Activity Promotion on Adiponectin, Leptin and Other Inflammatory Markers in Prediabetes: A Systematic Review and Meta-Analysis of Randomized Controlled Trials	Strengthen the evidence on the impact of physical activity promotion on inflammatory markers in prediabetes.	Meta-analysis found that physical activity with or without dietary or lifestyle modification reduces the level of leptin.	The role of physical activity in adipokines
Ghanbari-Niaki et al. (2010) [321]	Plasma Visfatin Is Increased after High-Intensity Exercise	Investigate the effects of repeated short bouts of high-intensity exercise on plasma visfatin and related metabolic responses	The elevation in plasma visfatin, together with increased plasma glucose and insulin concentrations immediately after high-intensity exercise	

## Data Availability

Not applicable.

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
