# Peer review of "The Role of Adipokines in Health and Disease"

_biomedicines, 2023, doi:10.3390/biomedicines11051290_

Round 1
Reviewer 1 Report
This manuscript covers wide spectrum of adipokines and health. The manucript is well organized and has useful information to the readers.
Unfortunately, this review is not far from the previous reviews on the similar theme. Nevertheless, common adipokines are well described and health status including inflammation, metabolic disorders, cancer, and microbiota is also mentioned.
Further studies may benefit the next review on this theme.
Author Response
We appreciate your words and the time you took to read our review. Some aspects have been improved at the request of other reviewers. Thank you again for your input
Reviewer 2 Report
The authors of the manuscript no. 2339017 subtitled "The role of adipokines in health and disease" presented extensive information regarding Adipokines In cardiovascular disease, atherosclerosis and cancer. Overall, the manuscript is well written, but I have a few questions and comments.
How will the authors explain the role of IL-6, which is produced by both beating and brown adipose tissue.
In addition, it is advisable to describe the figure ……1,2,3.
Line 419 The abbreviation was already given at the beginning Monocyte chemotactic protein-1
Line 816 The abbreviation was already given at the beginning… plasminogen activator inhibitor-1 (Pai-1)
Line 816. Left blank brackets …..pathology (). In this sense, Gonullu et al.[209] (reported
Conclusion /Conclusions need strengthening.
Reviewer 3 Report
Title: The role of adipokines in health and disease
Authors: Vicente Javier Clemente-Suárez, Laura Redondo-Flórez; Ana Isabel Beltrán-Velasco; Alexandra Martín-Rodríguez; Ismael Martínez-Guardado; Eduardo Navarro-Jiménez; Carmen Cecilia Laborde-Cárdenas and José Francisco Tornero-Aguilera
General comment:
Recent years have changed our understanding of adipose tissue from energy storage to an active endocrine organ that via secreted adipokines can impact every organ and tissue in the human body. In their paper, Vicente Javier Clemente-Suárez et al. attempted to review the role of adipokines in human health and disease. This aim is challenging given the plethora of data in the field.
Major revisions:
1) The reader has the feeling that particular sections have been written by different authors and finally combined in one review manuscript. Therefore, for instance, some data is repeated and the abbreviations are not used consistently. Therefore, the manuscript requires a profound reorganization to make all sections consistent and avoid repetitions.
For instance: Section 5. "Adipokines and cardiovascular disease" contains the following paragraph: “These cytokines are known as adipokines, which could be defined as different pro- and anti-inflammatory peptides secreted by adipose 364 tissue, being adiponectin, leptin, retinol-binding protein 4, resistin and chemerin those more relevant[50,73]. These substances have an important role in inflammation events, being crucial in different cellular signaling processes. Thus, their activities include energy balance and lipid metabolism modulation, immune response regulation, vascular homeostasis, and angiogenesis differentiation as well as increased insulin sensitivity and resistance [74,75].” – all this data has been already presented in the previous sections including Introduction and Adipokines and inflammation.
2) Section numbering also suggests that there was some kind of inconsistency among the authors – while in sections 3-9 and 11-12, the particular adipokines are described in separate paragraphs, in sections 10 and 12, each adipokine is described in a separate subsection.
3) Given the fact that adipose tissue dysfunction in the course of obesity is responsible for abnormal adipokine secretion and related health consequences, the pathomechanism of this phenomenon should be introduced.
4) Introduction - this section should briefly present the rationale for the review. Please make it more concise and avoid repetitions.
5) Methods. The Authors state that this review is a narrative one; however in the Methods section, they present some of the selection criteria for the references included in the manuscript. If the Authors decide to keep the Methods section it should be more informative and include the precise list of keywords used during the search in databases, the number of references found initially, and the number of papers finally included in the analysis.
6) Conclusions – this section should be developed and provided with for instance a table summarizing the chief findings coming from particular subsections.
Minor revisions:
1) Whole manuscript: please explain abbreviations as they only appear in the text e.g. “TNFα, MDD, NAFLD” and use the abbreviations consequently
2) Abstract:
Line 24: “For this aim the present review (…) eating metabolic, cancer, and eating behaviors; finally, the role of microbiota, nutrition, and physical activity in adipokines is discussed” – please consider removing the repetition (“eating”)
3) Keywords
Please consider adding “adipokines” to the list
4) Introduction:
Lines 45-46: “Adipokines include leptin, adiponectin, resistin, ghrelin, and many others.” – please consider that ghrelin is a gut hormone, not an adipokine.
Please consider removing repetitions:
Lines 43-44 “Adipokines are cytokines that regulate inflammation, metabolism, appetite, cardiovascular function, immunity, and other physiological processes." next
Lines 53-54 “These cytokines have diverse effects on metabolism, inflammation, immunity, cardiovascular function, and cancer as it will be shown further on in the review.”
Lines 78-79 “These alterations in the production of adipokines of an inflammatory nature mostly explain the appearance and epidemiology of Western diseases, including cancer and metabolic disorders[9]." – please consider changing "These" to "Therefore."
Lines 80-81 “Among these diseases are type 2 diabetes, insulin resistance, and obesity. These adipokines result in alteration of glucose…" – there is no link between the two sentences.
Lines 107-108 “For example, a diet high in saturated fats and simple sugars has been shown to increase the production of pro-inflammatory adipokines, such as resistance and IL-6…” – please change resistance to resistin.
5) Section 7. Adipokines and mental disease:
- please explain why in this section all adipokines' names are written with a capital letter.

Round 2
Reviewer 2 Report
The authors have corrected the suggested points. Accept in present form. Thank you
Reviewer 3 Report
I would like to express my gratitude for the opportunity to re-review the paper 'The role of adipokines in health and disease' by Vicente Javier Clemente-Suárez et al. As the authors have taken into account my comments on the structure and content of the paper, I believe that in its current form the manuscript is suitable for publication in the Biomedicines journal.